# Concept-level Debugging of Part-Prototype Networks

**Andrea Bontempelli**[1], **Stefano Teso**[2,1], **Katya Tentori**[2,3], **Fausto Giunchiglia**[1], **Andrea Passerini**[1]
[1] DISI    [2] CIMeC    [3] DIPSCO
University of Trento
`name.surname@unitn.it`

## Abstract

Part-prototype Networks (ProtoPNets) are concept-based classifiers designed to achieve the same performance as black-box models without compromising transparency. ProtoPNets compute predictions based on similarity to class-specific part-prototypes learned to recognize parts of training examples, making it easy to faithfully determine what examples are responsible for any target prediction and why. However, like other models, they are prone to picking up confounders and shortcuts from the data, thus suffering from compromised prediction accuracy and limited generalization. We propose ProtoPDebug, an effective concept-level debugger for ProtoPNets in which a human supervisor, guided by the model's explanations, supplies feedback in the form of what part-prototypes must be forgotten or kept, and the model is fine-tuned to align with this supervision. Our experimental evaluation shows that ProtoPDebug outperforms state-of-the-art debuggers for a fraction of the annotation cost. An online experiment with laypeople confirms the simplicity of the feedback requested to the users and the effectiveness of the collected feedback for learning confounder-free part-prototypes. ProtoPDebug is a promising tool for trustworthy interactive learning in critical applications, as suggested by a preliminary evaluation on a medical decision making task.

## 1 Introduction

Part-Prototype Networks (ProtoPNets) are "gray-box" image classifiers that combine the transparency of case-based reasoning with the flexibility of black-box neural networks (Chen et al., 2019). They compute predictions by matching the input image with a set of learned *part-prototypes* – i.e., prototypes capturing task-salient *elements* of the training images, like objects or parts thereof – and then making a decision based on the part-prototype activations only. What makes ProtoPNets appealing is that, despite performing comparably to more opaque predictors, they *explain* their own predictions in terms of relevant part-prototypes and of examples that these are sourced from. These explanations are – by design – more faithful than those extracted by post-hoc approaches (Dombrowski et al., 2019; Teso, 2019; Lakkaraju & Bastani, 2020; Sixt et al., 2020) and can effectively help stakeholders to simulate and anticipate the model's reasoning (Hase & Bansal, 2020).

Despite all these advantages, ProtoPNets are prone – like regular neural networks – to picking up confounders from the training data (e.g., class-correlated watermarks), thus suffering from compromised generalization and out-of-distribution performance (Lapuschkin et al., 2019; Geirhos et al., 2020). This occurs even with well-known data sets, as we will show, and it is especially alarming as it can impact high-stakes applications like COVID-19 diagnosis (DeGrave et al., 2021) and scientific analysis (Schramowski et al., 2020).

We tackle this issue by introducing ProtoPDebug, a simple but effective interactive debugger for ProtoPNets that leverages their case-based nature. ProtoPDebug builds on three key observations: (i) In ProtoPNets, confounders – for instance, textual meta-data in X-ray lung scans (DeGrave et al., 2021) and irrelevant patches of background sky or foliage (Xiao et al., 2020) – end up appearing as part-prototypes; (ii) Sufficiently expert and motivated users can easily indicate which part-prototypes are confounded by inspecting the model's explanations; (iii) Concept-level feedback of this kind is context-independent, and as such it generalizes across instances.

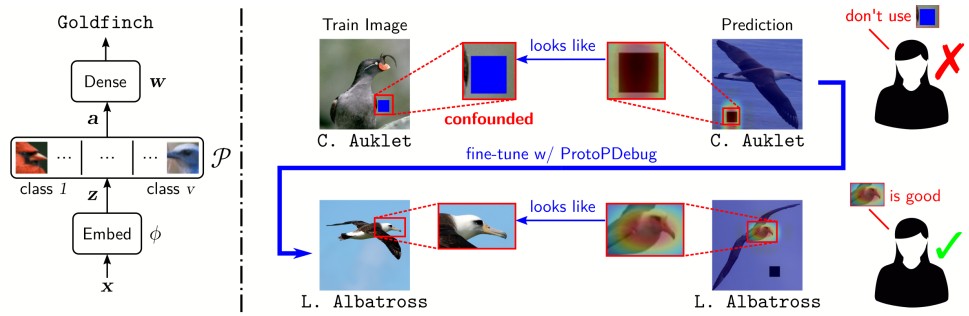

Figure 1: **Left**: architecture of ProtoPNets. **Right**: schematic illustration of the ProtoPDebug loop. The model has acquired a confounded part-prototype **p** (the blue square "■") that correlates with, but is not truly causal for, the `Crested Auklet` class, and hence mispredicts both unconfounded images of this class and confounded images of other classes (top row). Upon inspection, an end-user forbids the model to learn part-prototypes similar to **p**, achieving improved generalization (bottom row). Relevance of all part-prototypes is omitted for readability but assumed positive.

In short, ProtoPDebug leverages the explanations naturally output by ProtoPNets to acquire concept-level feedback about confounded (and optionally high-quality) part-prototypes, as illustrated in Fig. 1. Then, it aligns the model using a novel pair of losses that penalize part-prototypes for behaving similarly to confounded concepts, while encouraging the model to remember high-quality concepts, if any. ProtoPDebug is ideally suited for human-in-the-loop explanation-based debugging (Kulesza et al., 2015; Teso & Kersting, 2019), and achieves substantial savings in terms of annotation cost compared to alternatives based on input-level feedback (Barnett et al., 2021). In fact, in contrast to the per-pixel relevance masks used by other debugging strategies (Ross et al., 2017; Teso & Kersting, 2019; Plumb et al., 2020; Barnett et al., 2021), concept-level feedback automatically generalizes across instances, thus speeding up convergence and preventing relapse. Our experiments show that ProtoPDebug is effective at correcting existing bugs and at preventing new ones on both synthetic and real-world data, and that it needs less corrective supervision to do so than state-of-the-art alternatives.

**Contributions.** Summarizing, we: (1) Highlight limitations of existing debuggers for black-box models and ProtoPNets; (2) Introduce ProtoPDebug, a simple but effective strategy for debugging ProtoPNets that drives the model away from using confounded concepts and prevents forgetting well-behaved concepts; (3) Present an extensive empirical evaluation showcasing the potential of ProtoPDebug on both synthetic and real-world data sets.

## 2 PART-PROTOTYPE NETWORKS

ProtoPNets (Chen et al., 2019) classify images into one of $v$ classes using a three-stage process comprising an embedding stage, a part-prototype stage, and an aggregation stage; see Fig. 1 (left).

**Embedding stage:** Let **x** be an image of shape $w \times h \times d$, where $d$ is the number of channels. The embedding stage passes **x** through a sequence of (usually pre-trained) convolutional and pooling layers with parameters $\phi$, obtaining a latent representation $\mathbf{z} = h(\mathbf{x})$ of shape $w' \times h' \times d'$, where $w' < w$ and $h' < h$. Let $\mathcal{Q}(\mathbf{z})$ be the set of $1 \times 1 \times d'$ subtensors of **z**. Each such subtensor $\mathbf{q} \in \mathcal{Q}(\mathbf{z})$ encodes a filter in latent space and maps a rectangular region of the input image **x**.

**Part-prototype stage:** This stage memorizes and uses $k$ part-prototypes $\mathcal{P} = \{\mathbf{p}_1, \ldots, \mathbf{p}_k\}$. Each $\mathbf{p}_j$ is a tensor of shape $1 \times 1 \times d'$ explicitly learned – as explained below – so as to capture salient visual concepts appearing in the training images, like heads or wings. The activation of a part-prototype **p** on a part $\mathbf{q} \in \mathcal{Q}(\mathbf{z})$ is given by a *difference-of-logarithms* function, defined as (Chen et al., 2019):

$$\text{act}(\mathbf{p}, \mathbf{q}) := \log(\|\mathbf{p} - \mathbf{q}\|^2 + 1) - \log(\|\mathbf{p} - \mathbf{q}\|^2 + \epsilon) \geq 0 \quad (1)$$

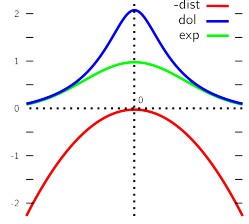

Figure 2: Part-prototype activation functions.

Here, $\|\cdot\|$ indicates the $L_2$ norm and $\epsilon > 0$ is a small constant. Alternatively, one can employ an *exponential* activation, defined as (Hase & Bansal, 2020):

$$\text{act}(\mathbf{p}, \mathbf{q}) = \exp(-\gamma \|\mathbf{p} - \mathbf{q}\|^2) \qquad (2)$$

Both activation functions are bell-shaped and decrease monotonically with the distance between $\mathbf{q}$ and $\mathbf{p}$, as illustrated in Fig. 2. The image-wide activation of $\mathbf{p}$ is obtained via max-pooling, i.e., $\text{act}(\mathbf{p}, \mathbf{z}) := \max_{\mathbf{q} \in \mathcal{Q}(\mathbf{z})} \text{act}(\mathbf{p}, \mathbf{q})$. Given an input image $\mathbf{x}$, this stage outputs an activation vector that captures how much each part-prototype activates on the image:

$$\mathbf{a}(\mathbf{x}) := (\text{act}(\mathbf{p}, h(\mathbf{x})) : \mathbf{p} \in \mathcal{P}) \in \mathbb{R}^k \qquad (3)$$

The activation vector essentially encodes the image according to the concept vocabulary. Each class $y \in [v] := \{1, \ldots, v\}$ is assigned $\lfloor \frac{k}{v} \rfloor$ discriminative part-prototypes $\mathcal{P}^y \subseteq \mathcal{P}$ learned so as to strongly activate (only) on training examples of that class, as described below.

**Aggregation stage:** In this stage, the network computes the score of class $y$ by aggregating the concept activations using a dense layer, i.e., $\sum_j w_j^y a_j(\mathbf{x})$, where $\mathbf{w}^y \in \mathbb{R}^k$ is the weight vector of class $y$, and applies a softmax function to obtain conditional probabilities $p_\theta(y \mid \mathbf{x})$. The set of parameters appearing in the ProtoPNet will be denoted by $\theta$.

**Loss and training.** The network is fit by minimizing a compound empirical loss over a data set $D$:

$$\ell(\theta) := -\frac{1}{|D|} \sum_{(\mathbf{x}, y) \in D} \log p_\theta(y \mid \mathbf{x}) + \lambda_c \ell_{\text{cls}}(\theta) + \lambda_s \ell_{\text{sep}}(\theta) \qquad (4)$$

This comprises a cross-entropy loss (first term) and two regularization terms that encourage the part-prototypes to cluster the training set in a discriminative manner:

$$\ell_{\text{cls}}(\theta) := \frac{1}{|D|} \sum_{(\mathbf{x}, y)} \min_{\substack{\mathbf{p} \in \mathcal{P}^y \\ \mathbf{q} \in \mathcal{Q}(h(\mathbf{x}))}} \|\mathbf{p} - \mathbf{q}\|^2 \quad (5) \qquad \ell_{\text{sep}}(\theta) := -\frac{1}{|D|} \sum_{(\mathbf{x}, y)} \min_{\substack{\mathbf{p} \in \mathcal{P} \setminus \mathcal{P}^y \\ \mathbf{q} \in \mathcal{Q}(h(\mathbf{x}))}} \|\mathbf{p} - \mathbf{q}\|^2 \quad (6)$$

Specifically, the part-prototypes are driven by the clustering loss $\ell_{\text{cls}}$ to cover all examples of their associated class and by the separation loss $\ell_{\text{sep}}$ *not* to activate on examples of other classes.

During training, the embedding and part-prototype layers are fit jointly, so as to encourage the data to be embedded in a way that facilitates clustering. At this time, the aggregation weights are fixed: each weight $w_j^u$ of $\mathbf{p}_j \in \mathcal{P}^y$ is set to 1 if $u = y$ and to $-0.5$ otherwise. The aggregation layer is fit in a second step by solving a logistic regression problem. ProtoPNets guarantee the part-prototypes to map to concrete cases by periodically projecting them onto the data. Specifically, each $\mathbf{p} \in \mathcal{P}^y$ is replaced with the (embedding of the) closest image part from any training example of class $y$.

**Explanations.** The architecture of ProtoPNets makes it straightforward to extract explanations highlighting, for each part-prototype $\mathbf{p}$: (1) its *relevance* for the target decision $(\mathbf{x}, y)$, given by the score $w_j a_j(\mathbf{x})$; (2) its *attribution map* $\text{attr}(\mathbf{p}, \mathbf{x})$, obtained by measuring the activation of $\mathbf{p}$ on each part $\mathbf{q}$ of $\mathbf{x}$, and then upscaling the resulting matrix to $w \times h$ using bilinear filtering:

$$\text{attr}(\mathbf{p}, \mathbf{x}) := \text{upscale}([\text{act}(\mathbf{p}, \mathbf{q})]_{\mathbf{q} \in \mathcal{Q}(\mathbf{x})}); \qquad (7)$$

(3) the *source example* that it projects onto.

## 3 INPUT-LEVEL DEBUGGING STRATEGIES AND THEIR LIMITATIONS

Explanations excel at exposing confounders picked up by models from data (Lapuschkin et al., 2019; Geirhos et al., 2020), hence constraining or supervising them can effectively dissuade the model from acquiring those confounders. This observation lies at the heart of recent approaches for debugging machine learning models, see (Teso et al., 2022) for an overview.

**A general recipe.** The general strategy is well illustrated by the right for the right reasons (RRR) loss (Ross et al., 2017), which aligns the attribution maps produced by a predictor to those supplied by a human annotator. Let $f : \mathbf{x} \mapsto y$ be a differentiable classifier and $IG(f, (\mathbf{x}, y)) \in \mathbb{R}^d$ be the input gradient of $f$ for a decision $(\mathbf{x}, y)$. This attribution algorithm (Baehrens et al., 2010; Simonyan et al., 2014) assigns relevance to each $x_j$ for the given decision based on the magnitude

of the gradient w.r.t. $x_j$ of the predicted probability of class $y$. The RRR loss penalizes $f$ for associating non-zero relevance to known irrelevant input variables:

$$\ell_{\mathrm{rrr}}(\theta) = \tfrac{1}{|D|} \sum_{(\mathbf{x},\mathbf{m},y)\in D} \|(1-\mathbf{m}) \odot IG(f,(\mathbf{x},y))\|^2 \tag{8}$$

Here, $\mathbf{m}$ is the ground-truth attribution mask of example $(\mathbf{x},y)$, i.e., $m_j$ is $0$ if the $j$-th pixel of $\mathbf{x}$ is *irrelevant* for predicting the ground-truth label and $1$ otherwise. Other recent approaches follow the same recipe (Rieger et al., 2020; Selvaraju et al., 2019; Shao et al., 2021; Viviano et al., 2021); see Lertvittayakumjorn & Toni (2021) and Friedrich et al. (2022) for an in-depth overview.

**Debugging ProtoPNets.** A natural strategy to debug a ProtoPNet with a confounded part-prototype $\mathbf{p}$ is to delete the latter and fine-tune the remaining part-prototypes $\mathcal{P}' := \mathcal{P} \setminus \{\mathbf{p}\}$. This solution is however very fragile, as nothing prevents the updated network from re-acquiring the same confounder using one of its remaining part-prototypes. One alternative is to freeze the part-prototypes $\mathcal{P}'$ and fine-tune the aggregation stage only, but this also problematic: in the presence of a strong confound, the model ends up relying chiefly on $\mathbf{p}$, with the remaining part-prototypes $\mathcal{P}'$ being low quality, not discriminative, and associated to low weights. Fine-tuning the aggregation layer on $\mathcal{P}'$ then typically yields poor prediction performance.

These issues prompted the development of the only existing debugger for ProtoPNets, IAIA-BL (Barnett et al., 2021). It also fits the above template, in that – rather than removing confounded part-prototypes – it penalizes those part-prototypes that activate on *pixels* annotated as irrelevant. The IAIA-BL loss is defined as:

$$\ell_{\mathrm{iaia}}(\theta) = \tfrac{\lambda_{\mathrm{iaia}}}{|D|} \sum_{(\mathbf{x},\mathbf{m},y)\in D} \left( \sum_{\mathbf{p}\in\mathcal{P}^y} \|(1-\mathbf{m}) \odot \mathrm{attr}(\mathbf{p},\mathbf{x})\| + \sum_{\mathbf{p}\in\mathcal{P}\setminus\mathcal{P}^y} \|\mathrm{attr}(\mathbf{p},\mathbf{x})\| \right) \tag{9}$$

where $\mathrm{attr}$ is the ProtoPNet's attribution map (see Eq. (7)) and $\odot$ indicates the element-wise product. The first inner summation is analogous to the RRR loss, while the second one encourages the part-prototypes of *other* classes not to activate at all, thus reinforcing the separation loss in Eq. (6).

**Limitations.** A critical issue with these approaches is that they are restricted to pixel-level supervision, which is inherently local: an attribution map that distinguishes between object and background in a given image does *not* generally carry over to other images. This entails that, in order to see any benefits for more complex neural nets, a substantial number of examples must be annotated individually, as shown by our experiments (see Fig. 3). This is especially troublesome as ground-truth attribution maps are not cheap to acquire, partly explaining why most data sets in the wild do not come with such annotations.

Put together, these issues make debugging with these approaches cumbersome, especially in human-in-the-loop applications where, in every debugging round, the annotator has to supply a non-negligible number of pixel-level annotations.

## 4   CONCEPT-LEVEL DEBUGGING WITH PROTOPDEBUG

Our key observation is that in ProtoPNets confounders only influence the output if they are recognized by the part-prototypes, and that therefore they can be corrected for by leveraging *concept-level supervision*, which brings a number of benefits. We illustrate the basic intuition with an example:

**Example:** *Consider the confounded bird classification task in Fig. 1 (right). Here, all training images of class* Crested Auklet *have been marked with a blue square "∎", luring the network into relying heavily on this confound, as shown by the misclassified* Laysian Albatro *image. By inspecting the learned part-prototypes – together with source images that they match strongly, for context – a sufficiently expert annotator can easily indicate the ones that activate on confounders, thus naturally providing concept-level feedback.*

Concept-level supervision sports several advantages over pixel-level annotations:

1. It is *cheap* to collect from human annotators, using an appropriate UI, by showing part-prototype activations on selected images and acquiring click-based feedback.
2. It is *very informative*, as it distinguishes between content and context and thus generalizes across instances. In our example, the blue square is a nuisance *regardless of what image it appears in*. In this sense, one concept-level annotation is equivalent to several input-level annotations.
3. By generalizing beyond individual images, it speeds up convergence and prevents relapse.

**The debugging loop.** Building on these observations, we develop ProtoPDebug, an effective and annotation efficient debugger for ProtoP-Nets that directly leverages concept-level supervision, which we describe next. The pseudocode for ProtoPDebug is listed in Algorithm 1. ProtoPDebug takes a ProtoPNet $f$ and its training set $D$ and runs for a variable number of debugging rounds. In each round, it iterates over all learned part-prototypes $\mathbf{p} \in \mathcal{P}$, and for each of them retrieves the $a$ training examples $(\mathbf{x}, y)$ that it activates the most on, including its source example. It then asks the user to judge $\mathbf{p}$ by inspecting its attribution map on each selected example. If the user indicates that a particular activation of $\mathbf{p}$ looks confounded, ProtoPDebug extracts a "cut-out" $\mathbf{x}_R$ of the confounder from the source image $\mathbf{x}$, defined as the box (or

> **Algorithm 1** A ProtoPDebug debugging session; $f$ is a ProtoPNet trained on data set $D$.
> ___
> 1: initialize $\mathcal{F} \leftarrow \varnothing, \mathcal{V} \leftarrow \varnothing$
> 2: **while** `True` **do**
> 3:     **for** $\mathbf{p} \in \mathcal{P}$ **do**
> 4:         **for** each $(\mathbf{x}, y)$ of the $a$ training examples most activated by $\mathbf{p}$ **do**
> 5:             **if** $\mathbf{p}$ looks confounded **then**
> 6:                 add cut-out $\mathbf{x}_R$ to $\mathcal{F}$
> 7:             **else if** $\mathbf{p}$ looks high-quality **then**
> 8:                 add cut-out $\mathbf{x}_R$ to $\mathcal{V}$
> 9:     **if** no confounders found **then**
> 10:         **break**
> 11:     fine-tune $f$ by minimizing $\ell(\theta) + \lambda_{\mathrm{f}}\ell_{\mathrm{for}}(\theta) + \lambda_{\mathrm{r}}\ell_{\mathrm{rem}}(\theta)$
> 12: **return** $f$

boxes, in case of disconnected activation areas) containing 95% of the part-prototype activation. It then embeds $\mathbf{x}_R$ using $h$ and adds it to a set of *forbidden concepts* $\mathcal{F}$. This set is organized into class-specific subsets $\mathcal{F}_y$, and the user can specify whether $\mathbf{x}_R$ should be added to the forbidden concepts for a specific class $y$ or to all of them (class-agnostic confounder). Conversely, if $\mathbf{p}$ looks particularly high-quality to the user, ProtoPDebug embeds its cut-out and adds the result to a set of *valid concepts* $\mathcal{V}$. This is especially important when confounders are ubiquitous, making it hard for the ProtoPNet to identify (and remember) non-confounded prototypes (see COVID experiments in Section 5.2). Importantly, we store the raw cut-out $\mathbf{x}_R$ rather than its embedding because the latter becomes obsolete upon fine-tuning.

Once all part-prototypes have been inspected, $f$ is updated so as to steer it away from the forbidden concepts in $\mathcal{F}$ while alleviating forgetting of the valid concepts in $\mathcal{V}$ using the procedure described below. Then, another debugging round begins. The procedure terminates when no confounder is identified by the user.

**Fine-tuning the network.** During fine-tuning, we search for updated parameters $\theta' = \{\phi', \mathcal{P}', \mathbf{w}'\}$ that are as close as possible to $\theta$ – thus retaining all useful information that $f$ has extracted from the data – while avoiding the bugs indicated by the annotator. This can be formalized as a *constrained distillation* problem (Gou et al., 2021):

$$\mathrm{argmin}_{\theta'} \ d(\theta', \theta)^2 \quad \text{s.t.} \quad \theta' \text{ activates little or not at all on cut-outs in } \mathcal{F} \qquad (10)$$

where $d$ is an appropriate distance function between sets of parameters. Since the order of part-prototypes is irrelevant, we define it as a permutation-invariant Euclidean distance:

$$d(\theta, \theta')^2 = \|\phi - \phi'\|^2 + \min_\pi \sum_{j \in [k]} \|\mathbf{p}_j - \mathbf{p}'_{\pi(j)}\|^2 + \|\mathbf{w} - \mathbf{w}'\|^2 \qquad (11)$$

Here, $\pi$ simply reorders the part-prototypes of the buggy and updated models so as to maximize their alignment. Recall that we are interested in correcting the concepts, so we focus on the middle term. Notice that the logarithmic and exponential activation functions in Eqs. (1) and (2) are both inversely proportional to $\|\mathbf{p} - \mathbf{p}'\|^2$ and achieve their maximum when the distance is zero, hence minimizing the distance is analogous to maximizing the activation. This motivates us to introduce two new penalty terms:

$$\ell_{\mathrm{for}}(\theta) := \frac{1}{v} \sum_{\substack{y \in [v]}} \max_{\substack{\mathbf{p} \in \mathcal{P}^y \\ \mathbf{f} \in \mathcal{F}_y}} \mathrm{act}(\mathbf{p}, \mathbf{f}) \qquad (12) \qquad \ell_{\mathrm{rem}}(\theta) := -\frac{1}{v} \sum_{\substack{y \in [v]}} \min_{\substack{\mathbf{p} \in \mathcal{P}_y \\ \mathbf{v} \in \mathcal{V}_y}} \mathrm{act}(\mathbf{p}, \mathbf{v}) \qquad (13)$$

The hard constraint in Eq. (10) complicates optimization, so we replace it with a smoother penalty function, obtaining a relaxed formulation $\mathrm{argmin}_{\theta'} \ d(\theta', \theta)^2 + \lambda \cdot \ell_{\mathrm{for}}(\theta)$. The *forgetting loss* $\ell_{\mathrm{for}}$ minimizes how much the part-prototypes of each class $y \in [v]$ activate on the most activated concept to be forgotten for that class, written $\mathcal{F}_y$. Conversely, the *remembering loss* $\ell_{\mathrm{rem}}$ maximizes how much the part-prototypes activate on the least activated concept to be remembered for that class, denoted $\mathcal{V}_y$. The overall loss used by ProtoPDebug for fine-tuning the model is then a weighted combination of the ProtoPNet loss in Eq. (4) and the two new losses, namely $\ell(\theta) + \lambda_{\mathrm{f}}\ell_{\mathrm{for}}(\theta) +$

Figure 3: Comparison between ProtoPDebug (in **red**), ProtoPNets on confounded data (in **black**), ProtoPNets on unconfounded data (**green**), and IAIA-BL with varying amount of attribution-map supervision (shades of **blue**) on the CUB5$_{box}$ data set. **Left to right**: micro $F_1$ on the training set, cross-entropy loss on the training set, and $F_1$ on the test set. Bars indicate std. error over 17 runs.

$\lambda_r \ell_{rem}(\theta)$, where $\lambda_f$ and $\lambda_r$ are hyper-parameters. We analyze the relationship between Eqs. (12) and (13) and other losses used in the literature in Appendix A.

**Benefits and limitations.** The key feature of ProtoPDebug is that it leverages concept-level supervision, which sports improved generalization across instances, cutting annotation costs and facilitating interactive debugging. ProtoPDebug naturally accommodates concept-level feedback that applies to specific classes. E.g., `snow` being useful for some classes (say, `winter`) but irrelevant for others (like `dog` or `wolf`) (Ribeiro et al., 2016). However, ProtoPDebug makes it easy to penalize part-prototypes of *all* classes for activating on class-agnostic confounders like, e.g., image artifacts. ProtoPDebug's two losses bring additional, non-obvious benefits. The remembering loss helps to prevent catastrophic forgetting during sequential debugging sessions, while the forgetting loss prevents the model from re-learning the same confounder in the future.

One source of concern is that, if the concepts acquired by the model are not understandable, debugging may become challenging. ProtoPNets already address this issue through the projection step, but following Koh et al. (2020); Marconato et al. (2022), one could guide the network toward learning a set of desirable concepts by supplying additional concept-level supervision and appropriate regularization terms. ProtoPDebug focuses on confounds that can be visually identified in part-prototypes, as this can facilitate interaction with the user. Confounds that do not affect the part-prototypes might also occur, but they are also less likely to affect the predictions. More generally, like other explanation-based methods, in sensitive prediction tasks ProtoPDebug poses the risk of leaking private information. Moreover, malicious annotators may supply adversarial supervision, corrupting the model. These issues can be avoided by restricting access to ProtoPDebug to trusted annotators only.

## 5 Empirical Analysis

In this section, we report results showing how concept-level supervision enables ProtoPDebug to debug ProtoPNets better than the state-of-the-art debugger IAIA-BL and using less and cheaper supervision, and that it is effective in both synthetic and real-world debugging tasks. ProtoPDebug and IAIA-BL were implemented on top of ProtoPNets (Chen et al., 2019) using the Adam optimizer (Kingma & Ba, 2014).

### 5.1 Concept-level vs instance-level debugging

We first compare the effectiveness of concept-level and instance-level supervision in a controlled scenario where the role of the confounders is substantial and precisely defined at design stage. To this end, we modified the CUB200 data set (Wah et al., 2011), which has been extensively used for evaluating ProtoPNets (Chen et al., 2019; Hase & Bansal, 2020; Rymarczyk et al., 2021; Hoffmann et al., 2021; Nauta et al., 2021). This data set contains images of 200 bird species in natural environments, with approximately 40 train and 20 test images per class. We selected the first five classes, and artificially injected simple confounders in the training images for three of them. The confounder is a colored square of fixed size, and it is placed at a random position. The color of the square is different for different classes, but it is absent from the test set, and thus acts as a perfect confounder for these three classes. The resulting synthetic dataset is denoted CUB5$_{box}$.

**Competitors and implementation details.** We compare ProtoPDebug, with supervision on *a single instance per confounder*, against the following competitors: 1) vanilla ProtoPNets; 2) a vanilla ProtoPNet trained on a clean version of the dataset without the confounders, denoted ProtoPNets$_{clean}$; 3) IAIA-BL $X$, the IAIA-BL model fit using ground-truth attribution maps on $X\%$ of the training examples, for $X \in \{5, 20, 100\}$, and IAIA-BL $n = 3$, with one attribution mask per confound (same number given to ProtoPDebug). Note that This setting is not interactive, but supervision is made available at the beginning of training. The embedding layers were implemented using a pre-trained VGG-16, allocating two prototypes for each class. The values of $\lambda_{\mathrm{f}}$ and $a$ were set to 100 and 5, respectively, and $\lambda_{\mathrm{iaia}}$ to 0.001, as in the original paper (Barnett et al., 2021); experiments with different values of $\lambda_{\mathrm{iaia}}$ gave inconsistent results. In this setting, all supervision is made available to all methods from the beginning, and the corrective feedback fed to the forgetting loss is class-specific.

**Results**. Fig. 3 reports the experimental results on all methods averaged over 15 runs. The left and middle plots show the training set macro $F_1$ and cross entropy, respectively. After 15 epochs, all methods manage to achieve close to perfect $F_1$ and similar cross entropy. The right plot shows macro $F_1$ on the test set. As expected, the impact of the confounders is rather disruptive, as can been seen from the difference between ProtoPNets and ProtoPNets$_{clean}$. Indeed, ProtoPNets end up learning confounders whenever present. Instance-level debugging seems unable to completely fix the problem. With the same amount of supervision as ProtoPDebug, IAIA-BL $n = 3$ does not manage to improve over the ProtoPNets baseline. Increasing the amount of supervision does improve its performance, but even with supervision on all examples IAIA-BL still fails to match the confound-free baseline ProtoPNets$_{clean}$ (IAIA-BL 100 curve). On the other hand, ProtoPDebug succeeds in avoiding to be fooled by confounders, reaching the performance of ProtoPNets$_{clean}$ despite only receiving a fraction of the supervision of the IAIA-BL alternatives. Note however that ProtoPDebug does occasionally select a *natural* confounder (the water), that is not listed in the set of forbidden ones. In the following section we show how to deal with unknown confounders via the full interactive process of ProtoPDebug.

## 5.2 PROTOPDEBUG IN THE REAL WORLD

**CUB5$_{nat}$ data set.** Next, we evaluate ProtoPDebug in a real world setting in which confounders occur naturally in the data as, e.g., background patches of sky or sea, and emerge at different stages of training, possibly as a consequence of previous user corrections. To maximize the impact of the natural confounders, we selected the 20 CUB200 classes with largest test $F_1$ difference when training ProtoPNets only on birds (removing background) or on entire images (i.e., bird + background), and testing them on images where the background has been shuffled among classes. Then, we applied ProtoPDebug focusing on the five most confounded classes out of these 20. We call this dataset CUB5$_{nat}$. All architectures and hyperparameters are as before.

**Experimental Protocol.** To evaluate the usability and effectiveness of ProtoPDebug with real users, we ran an online experiment, through Prolific Academic (prolific.com), one of the most reliable crowdsourcing platforms for behavioral research. Participants were laypeople, all native speakers of English. To make the experimental stimuli as clear as possible, separate patches for the activation area of the prototype on an image were automatically extracted from the 5 most-activate images ($a = 5$). See Appendix H for the detailed procedure. For each debugging round, a new group of 10 participants was asked to inspect the images and to determine whether the highlighted patch covered some part of the bird or whether, instead, it covered exclusively (or very nearly so) the background. A patch was flagged as a confounder when 80% of the users indicated it as covering the background. A prototype was accepted when no confounder was identified in its most activated image. Regardless of this, confounders in less activated images of the prototype were still collected as they might serve as feedback for other prototypes (class-agnostic confounders) in possible following iteration(s). The procedure ended when all prototypes were accepted.

**Results.** The average evaluation time for each image was 6 seconds. Across all images, participants' inter-annotation agreement was 96%, confirming the simplicity of the users' feedback required by ProtoPDebug. Fig. 4 shows the part-prototypes progressively learned by ProtoPDebug. In the first iteration, when no corrective feedback has been provided yet, the model learns confounders for most classes: the branches for the second and the last classes, the sky for the third one. Upon receiving user feedback, ProtoPDebug does manage to avoid learning the same confounders again in most cases, but it needs to face novel confounders that also correlate with the class (i.e., the sea for the

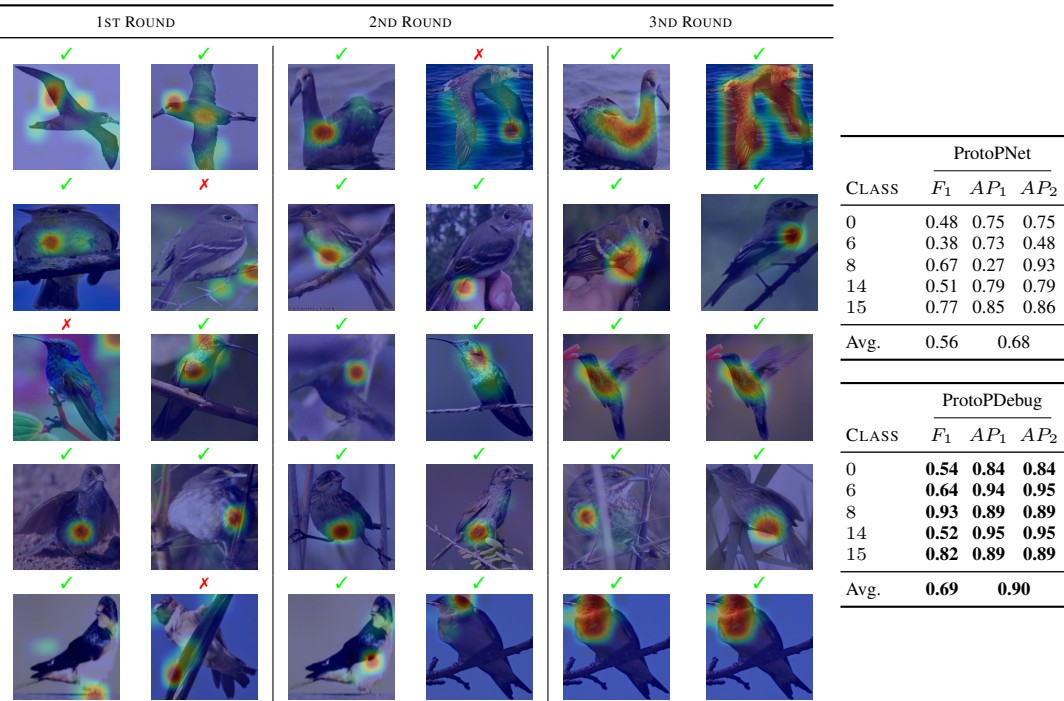

| | | ProtoPNet | |
|---|---|---|---|
| CLASS | $F_1$ | $AP_1$ | $AP_2$ |
| 0 | 0.48 | 0.75 | 0.75 |
| 6 | 0.38 | 0.73 | 0.48 |
| 8 | 0.67 | 0.27 | 0.93 |
| 14 | 0.51 | 0.79 | 0.79 |
| 15 | 0.77 | 0.85 | 0.86 |
| Avg. | 0.56 | 0.68 | |

| | | ProtoPDebug | |
|---|---|---|---|
| CLASS | $F_1$ | $AP_1$ | $AP_2$ |
| 0 | **0.54** | **0.84** | **0.84** |
| 6 | **0.64** | **0.94** | **0.95** |
| 8 | **0.93** | **0.89** | **0.89** |
| 14 | **0.52** | **0.95** | **0.95** |
| 15 | **0.82** | **0.89** | **0.89** |
| Avg. | **0.69** | **0.90** | |

Figure 4: **Left**: Three rounds of sequential debugging with ProtoPDebug on CUB5$_{nat}$. Rows: part-prototypes (two per class) and user feedback (checkmark vs. cross). Note that the prototypes produced before the first correction (left-most column) correspond to those learned by plain ProtoPNets. **Right**: Test set performance of ProtoPNets and ProtoPDebug on CUB5$_{nat}$. $AP_i$ is the attribution precision of $i$-th part-prototype. Bold denotes better performance.

first class). After two rounds, all prototypes are accepted, and the procedure ends. Fig. 4 reports the test set performance of ProtoPNet compared to ProtoPDebug in terms of $F_1$ and interpretability metric. The latter has been introduced in (Barnett et al., 2021, Eq. 6) to quantify the portion of pixels that are activated by a prototype. We slightly modified it to consider also their activation value (see Suppl. for the definition). Results show that ProtoPDebug improves over ProtoPNets in all classes (with an increase in $F_1$ of up to 0.26 for classes 6 and 8), and that it manages to learn substantially more interpretable prototypes (i.e., it is right for the right reason (Schramowski et al., 2020)), with a 22% improvement in activation precision on average.

**COVID data set.** The potential of artificial intelligence and especially deep learning in medicine is huge (EJ., 2019). On the other hand, the medical domain is known to be badly affected by the presence of confounders (Jager et al., 2008; Smith & Nichols, 2018). In order to evaluate the potential of ProtoPDebug to help addressing this issue, we tested it on a challenging real-world problem from the medical imaging domain. The task is to recognize COVID-19 from chest radiographies. As shown in DeGrave et al. (2021), a classifier trained on this dataset heavily relies on confounders that correlate with the presence or absence of COVID. These confounders come from the image acquisition procedure or annotations on the image. We trained and tested ProtoPDebug on the same datasets used in DeGrave et al. (2021) using the data pipeline source code (cod, 2021a). To simplify the identification of confounders, we focused on a binary classification task, discriminating COVID-positive images from images without any pathology. Given the extremely high inter-annotator agreement of the CUB5$_{nat}$ experiment, this experiment was conducted internally in our lab. We leave a crowdsourced evaluation to an extended version of this work.

**Results**. Fig. 5 reports the (non-zero activation) prototypes of ProtoPDebug at different correction rounds. As for Fig. 4 (left), the left-most column corresponds to the prototypes learned by ProtoPNets. Note that for each prototype, the supervision is given to the 10 most-activated images ($a = 10$). Penalized confounders have been extracted from images on which the prototype has non-zero activation, because they influence the classification. However, the patches of the images to remember are extracted even if the activation is zero, in order to force the prototype to increase

| | 1ST ROUND | | 2ND ROUND | | 3ND ROUND | | 4TH ROUND | |
|---|---|---|---|---|---|---|---|---|

Figure 5: Four rounds of sequential debugging with ProtoPDebug on COVID. Only the prototypes with non-zero activation are reported: first prototype refers to COVID- and second COVID+.

the activation on them thanks to the remembering loss. Eventually, ProtoPDebug manages to learn non-confounded prototypes, resulting in substantially improved test classification performance. The test $F_1$ goes from 0.26 of ProtoPNets (first column) to 0.54 at the end of the debugging process.

## 6 RELATED WORK

**Input-level debuggers.** ProtoPDebug is inspired by approaches for explanatory debugging (Kulesza et al., 2015) and explanatory interactive learning (Teso & Kersting, 2019; Schramowski et al., 2020), which inject explanations into interactive learning, enabling a human-in-the-loop to identify bugs in the model's reasoning and fix them by supplying corrective feedback. They leverage input attributions (Teso & Kersting, 2019; Selvaraju et al., 2019; Lertvittayakumjorn et al., 2020; Schramowski et al., 2020), example attributions (Teso et al., 2021; Zylberajch et al., 2021), and rules (Popordanoska et al., 2020), but are not designed for concept-based models nor leverage concept-level explanations. IAIA-BL (Barnett et al., 2021) adapts these strategies to PPNets by penalizing part-prototypes that activate on irrelevant regions of the input, but it is restricted to instance-level feedback, which is neiter as cheap to obtain nor as effective as concept-level feedback, as shown by our experiments. For an overview of interactive debugging strategies, see Teso et al. (2022).

**Concept-level debuggers.** Stammer et al. (2021) debug neuro-symbolic models by enforcing logical constraints on the concepts attended to by a network's slot attention module (Locatello et al., 2020), but requires the concept vocabulary to be given and fixed. ProtoPDebug has no such requirement, and the idea of injecting prior knowledge (orthogonal to our contributions) could be fruitfully integrated with it. ProSeNets (Ming et al., 2019) learn (full) prototypes in the embedding space given by a sequence encoder and enable users to steer the model by adding, removing and manipulating the learned prototypes. The update step however requires input-level supervision and adapts the embedding layers only. iProto-TREX (Schramowski et al., 2021) combines ProtoPNets with transformers, and learns part-prototypes capturing task-relevant text snippets akin to rationales (Zaidan et al., 2007). It supports dynamically removing and hot-swapping bad part-prototypes, but it lacks a forgetting loss and hence is prone to relapse. Applying ProtoPDebug to iProto-TREX is straightforward and would fix these issues. Bontempelli et al. (2021) proposed a classification of debugging strategies for concept-based models, but presented no experimental evaluation. Lage & Doshi-Velez (2020) acquire concepts by eliciting concept-attribute dependency information with questions like "does the concept depression depend on feature lorazepam?". This approach can prevent confounding, but it is restricted to white-box models and interaction with domain experts. FIND (Lertvittayakumjorn et al., 2020) offers similar functionality for deep NLP models, but it relies on disabling concepts only.

**Other concept-based models.** There exist several concept-based models (CBMs) besides ProtoP-Nets, including self-explainable neural networks (Alvarez-Melis & Jaakkola, 2018), concept bottleneck models (Koh et al., 2020; Losch et al., 2019), and concept whitening (Chen et al., 2020). Like ProtoPNets, these models stack a simulatable classifier on top of (non-prototypical) interpretable concepts. Closer to our setting, prototype classification networks (Li et al., 2018) and deep embedded prototype networks (Davoudi & Komeili, 2021) make predictions based on embedding-space prototypes of full training examples (rather than parts thereof). Our work analyzes ProtoPNets as they perform comparably to other CBMs while sporting improved interpretability (Chen et al., 2019; Hase & Bansal, 2020). Several extensions of ProtoPNets have also been proposed (Hase et al., 2019; Rymarczyk et al., 2021; Nauta et al., 2021; Kraft et al., 2021). ProtoPDebug naturally applies to these variants, and could be extended to other CBMs by adapting the source example selection and update steps. An evaluation on these extensions is left to future work.

**Ethics statement** ProtoPDebug is conceived for leveraging interactive human feedback to improve the trustworthiness and reliability of machine predictions, which is especially important for critical applications like medical decision making. On the other hand, adversarial annotators might supply corrupted supervisions to alter the model and drive its predictions towards malignant goals. For these reasons, access to the model and to the explanations should be restricted to authorized annotators only, as standard in safety-critical applications. Concerning the online experiment with laypeople, all participants were at least 18 years old and provided informed consent prior to participating in the research activities.

**Reproducibility statement** All experiments were implemented in Python 3 using Pytorch (Paszke et al., 2019) and run on a machine with two Quadro RTX 5000 GPUs. Each run requires in-between 5 and 30 minutes. We implemented ProtoPDebug and IAIA-BL on top of ProtoPNets and we used the source code of the respective authors of the two methods (cod, 2019; 2021b). The full experimental setup is published on GitHub at `https://github.com/abonte/protopdebug` and in the Supplementary Material together with additional implementation details. The repository contains the configuration files reporting the hyper-parameters of all experiments, and the procedures to process the data and reproduce the experiments. The implementation details are also available in Appendix C, while Appendix D informs about the data sets, which are all made freely available by their respective authors. For CUB200 (Wah et al., 2011) and image masks (Farrell, 2022), the image pre-processing reuses the code of the ProtoPNets authors (cod, 2019). The COVID images are processed using (cod, 2021a). The images are downloaded from GitHub-COVID repository (Cohen et al., 2020), ChestX-ray14 repository (Wang et al., 2017), PadChest (Bustos et al., 2020) and BIMCV-COVID19+ (Vayá et al., 2020). Appendix H provides the experimental protocol for the online experiment with real users.

ACKNOWLEDGMENTS

This research of FG has received funding from the European Union's Horizon 2020 FET Proactive project "WeNet - The Internet of us", grant agreement No. 823783, The research of AB was supported by the "DELPhi - DiscovEring Life Patterns" project funded by the MIUR Progetti di Ricerca di Rilevante Interesse Nazionale (PRIN) 2017 – DD n. 1062 del 31.05.2019. The research of ST and AP was partially supported by TAILOR, a project funded by EU Horizon 2020 research and innovation programme under GA No 952215.

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

## A    RELATIONSHIP TO OTHER LOSSES

The forgetting loss in Eq. (12) can be viewed as a *lifted* version of the separation loss in Eq. (6): the former runs over classes and leverages per-class concept feedback, while the latter runs over examples. The same analogy holds for the remembering and clustering losses, cf. Eqs. (5) and (13). The other difference is that the instance-level losses are proportional to the negative squared distance $-\|\mathbf{p} - \mathbf{q}\|^2$ between part-prototypes and parts and the concept-level ones to the activation instead. However, existing activation functions are monotonically decreasing in the squared distance, meaning that they only really differ in fall-off speed, as can be seen in Fig. 2. The IAIA-BL loss can also be viewed as a (lower bound of the) separation loss. Indeed:

$$\ell_{\text{iaia}}(\theta) = \sum_{(\mathbf{x},\mathbf{m},y)} \sum_{\mathbf{p} \in \mathcal{P}^y} \|(1 - \mathbf{m}) \odot \text{attr}(\mathbf{p}, \mathbf{x})\| \leq \sqrt{k} \sum_{(\mathbf{x},\mathbf{m},y)} \max_{\mathbf{p} \in \mathcal{P}^y} \|(1 - \mathbf{m}) \odot \text{attr}(\mathbf{p}, \mathbf{x})\|_\infty$$

$$= \sqrt{k} \sum_{(\mathbf{x},\mathbf{m},y)} \max_{\substack{\mathbf{p} \in \mathcal{P}^y \\ \mathbf{q} \in \mathcal{Q}(h(\mathbf{x}))}} |(1 - \overline{m}_{\mathbf{q}}) \cdot \text{act}(\mathbf{p}, \mathbf{q}))| = -\sqrt{k} \sum_{(\mathbf{x},\mathbf{m},y)} \min_{\mathbf{p},\mathbf{q}} (1 - \overline{m}_{\mathbf{q}}) \cdot \|\mathbf{p} - \mathbf{q}\|^2$$

where $\overline{m}_{\mathbf{q}} \geq 0$ is the "part-level" attribution mask for part $\mathbf{q}$, obtained by down-scaling the input-level attribution mask of all pixels in the receptive field of $\mathbf{q}$. This formulation highlights similarities and differences with our forgetting loss: the latter imposes similar constraints as the IAIA-BL penalty, but (*i*) it does not require full mask supervision, and (*ii*) it can easily accommodate concept-level feedback that targets entire *sets* of examples, as explained above.

## B    ACTIVATION PRECISION

In order to measure the quality of the explanations produced by the various models, we adapted the *activation precision* (AP) metric from the original IAIA-BL experiments (Barnett et al., 2021, Eq. 6). Assuming that the attribution mask attr output by ProtoPNets is flattened to a vector with the same shape as $\mathbf{m}$, the original formulation of AP can be written as:

$$\frac{1}{|D|} \sum_{(\mathbf{x},\mathbf{m},y) \in D} \left( \frac{\sum_i [\mathbf{m} \odot T_\tau(\text{attr}(\mathbf{p}, \mathbf{x}))]_i}{\sum_i T_\tau(\text{attr}(\mathbf{p}, \mathbf{x}))_i} \right) \tag{14}$$

Here, $T_\tau$ is a threshold function that takes an attribution map (i.e., a real-valued matrix), computes the $(100 - \tau)$ percentile of the attribution values, and sets the elements above this threshold to 1 and the rest to 0. Intuitively, the AP counts how many pixels predicted as relevant by the network are actually relevant according to the ground-truth attribution mask $\mathbf{m}$.

This definition however ignores differences in activation across the most relevant pixels, which can be quite substantial, penalizing debugging strategies that manage to correctly shift *most* of the network's explanations over the truly relevant regions. To fix this, we modified $T_\tau$ not to discard this information, as follows:

$$\frac{1}{|D|} \sum_{(\mathbf{x},\mathbf{m},y) \in D} \left( \frac{\sum_i [\mathbf{m} \odot T_\tau(\text{attr}(\mathbf{p}, \mathbf{x})) \odot \text{attr}(\mathbf{p}, \mathbf{x})]_i}{\sum_i [T_\tau(\text{attr}(\mathbf{p}, \mathbf{x})) \odot \text{attr}(\mathbf{p}, \mathbf{x})]_i} \right) \tag{15}$$

In our experiments, we set $\tau = 5\%$. The same value is used by IAIA-BL and ProtoPNets for visualizing the model's explanations and is therefore what the user would see during interaction.

## C   IMPLEMENTATION DETAILS

**Architectures.**   We use the same ProtoPNet architecture for all competitors:

- *Embedding step*: For the CUB experiments, the embedding layers were taken from a pre-trained VGG-16, as in (Chen et al., 2019), and for COVID from a VGG-19 (Simonyan & Zisserman, 2014).
- *Adaptation layers*: As in (Chen et al., 2019), two $1 \times 1$ convolutional layers follow the embedding layers, and use ReLu and sigmoid activation functions respectively. The output dimension is $7 \times 7$.
- *Part-prototype stage*: All models allocate exactly two part-prototypes per class, as allocating more did not bring any benefits. The prototypes are tensors of shape $1 \times 1 \times 128$.

To stabilize the clustering and separation losses, we implemented two "robust" variants proposed by Barnett et al. (2021), which are obtained by modifying the regular losses in Eqs. 5 and 6 of the main text to consider the average distance to the closest 3 part-prototypes, rather than the distance to the closest one only. These two changes improved the performance of all competitors.

**Training.**   We froze the embedding layer so as to drive the model toward updating the part-prototypes. Similarly to the ProtoPNet implementation, the learning rate of the prototype layer was scaled by $0.15$ every $4$ epochs. The training batch size is set to 20 for the experiments on $CUB5_{box}$ and COVID data sets, and 128 on $CUB5_{nat}$.

We noticed that, in confounded tasks, projecting the part-prototypes onto the nearest latent training patch tends the prototypes closer to confounders, encouraging the model to rely on it. This strongly biases the results *against* the ProtoPNet baseline. To avoid this problem, in our experiments we disabled the projection step.

**Hyper-parameters.**   The width of the activation function (Eq. 1 in the main paper) used in $\ell_{for}$ was set to $\epsilon = 10^{-8}$ in all experiments.

For $CUB5_{box}$ and $CUB5_{nat}$, the weights of the different losses were set to $\lambda_{cls} = 0.5$, $\lambda_{sep} = 0.08$, $\lambda_{for} = 100$. In this experiment, the remembering loss was not necessary and therefore it was disabled by setting $\lambda_{rem} = 0$. The value of $\lambda_{iaia} = 0.001$ was selected from $\{0.001, 0.01, 1.0, 100\}$ so as to optimize the $F_1$ test set performance, averaged over three runs. The number of retrieved training examples shown to the user is 5 ($a = 5$).

For the COVID data set, $\lambda_{cls}$, $\lambda_{sep}$, $\lambda_{for}$ and $\lambda_{rem}$ were set to $0.5, 0.08, 200$, and $0$ respectively for the first two debugging rounds. In the last round, $\lambda_{cls}$, $\lambda_{sep}$ were decreased to $25\%$ of the above values and $\lambda_{rem}$ increased to $0.01$, to boost the effect of $\ell_{for}$ and $\ell_{rem}$. This shifts the focus of training from avoiding very bad part-prototypes, to adjusting the partly wrong prototypes and increasing the activation on the relevant patches. The value of $a$ is set to 10.

## D   DATA STATISTICS AND PREPARATION

Table 1 reports statistics for all data sets used in our experiments, including the number of training and test examples, training examples used for the visualization step, and classes. The visualization data set contains the original images, i.e., before the augmentation step, and is used to visualize the images on which the prototypes activate most during the debugging loop. All images were resized to $224 \times 224 \times 3$, as done in ProtoPNet (Chen et al., 2019), IAIA-BL (Barnett et al., 2021) and for the COVID-19 data in (DeGrave et al., 2021).

### D.1   CUB DATA SETS

Classes in the CUB200 data set include only 30 training examples each. To improve the performance of ProtoPNets, as done in (Chen et al., 2019), we augmented the training images using random rotations, skewing, and shearing, resulting in 900 augmented examples per class.

**$CUB5_{box}$.**   The five CUB200 classes that were make up the $CUB5_{box}$ data set are listed Fig. 7 (left). Synthetic confounders (i.e., colored boxes) have been added to *training* images of the first,

Table 1: Statistics for all data sets used in our experiments.

| DATA SET | # TRAIN | # TEST | # VISUALIZATION | # OF CLASSES |
|---|---|---|---|---|
| CUB5$_{box}$ | 4500 | 132 | 150 | 5 |
| CUB5$_{nat}$ | 180000 | 569 | 600 | 20 |
| COVID | 7640 | 1744 | 7640 | 2 |

Figure 6: From left to right, the images of confounders on which $\ell_{\text{for}}$ relies for computing the activation value: the confounding green box present in CUB5$_{box}$ and two background patches in CUB5$_{nat}$.

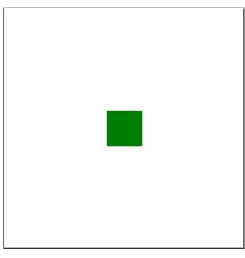
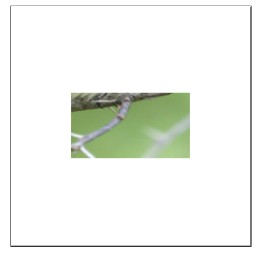
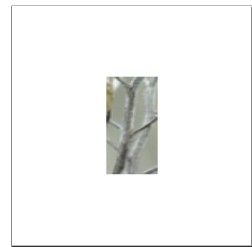

second and last class. No confounder was added to the other two classes. Fig. 6 (left) shows one of the three colored boxes. An example image confounded with a green box "■" is shown in Fig. 7 (right) along with the corresponding ground-truth attribution mask **m** used for IAIA-BL.

Figure 7: **Left**: List of CUB200 classes selected for the CUB5$_{box}$ data set. **Right**: Confounded training images from CUB5$_{box}$ and corresponding ground-truth attribution mask.

| # | CLASS | LABEL | +CONFOUND? |
|---|---|---|---|
| 0 | 001 | Black-footed Albatross | ✓ |
| 1 | 002 | Laysan Albatross | ✓ |
| 2 | 003 | Sooty Albatross | ✗ |
| 3 | 004 | Groove-billed Ani | ✗ |
| 4 | 005 | Crested Auklet | ✓ |

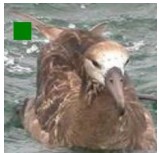
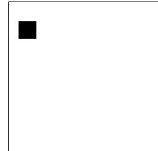

**CUB5$_{nat}$.** We created a variation of CUB200, namely CUB5$_{nat}$, to introduce a confounder that occurs naturally, e.g., sky or trees patches in the background, and are learned during the training phase. We followed these steps to select the twenty classes reported in Table 2 and used for the experiments:

1. We modified the *test set* images of all the 200 classes of CUB200 by placing the bird of one class on the background of another one. Since the outline of the bird removed from the background may not match the pasted one, the uncovered pixels are set to random values. Figure Fig. 8 shows the background swap for one image and the corresponding ground-truth mask. This background shift causes a drop in the performance of the model that relies on the background to classify a bird.

2. We trained ProtoPNet on two variations of the *training set*: in the first, by removing the background, the images contain only the birds, and in the second, both the bird and the background. We selected the twenty classes with the maximum gap in terms of $F_1$ on the modified test set, out of the 200 classes of CUB200. Thus, the background appears to be a confounder for these twenty classes. The right most images in Fig. 6 highlight patches of the background learned by the model.

3. We repeated the step above but only on the twenty classes, and we selected the five classes with the maximum gap. In the experiments, we provided supervision only on these five classes. Since the experiments are run on this subset of twenty classes, the separation loss might force the model to learn different prototypes with respect to the ones learned when

Figure 8: From left to right: original image with the sea in the background, the same bird but with the background of the land and the corresponding ground-truth attribution mask.

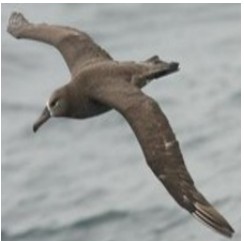 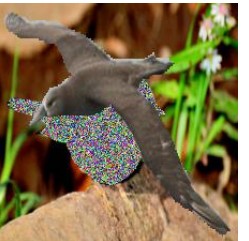 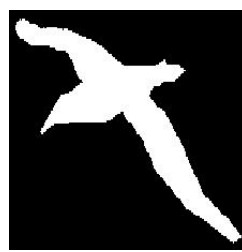

trained on 200 classes. Hence, the prototypes activate on different pixels and the classes with the largest gap are different. This additional ranking of the classes allows us to debug the most problematic ones and to show the potential performance improvement on them. For this reason, the results reported in the main paper are computed on only these five classes.

Table 2: Twenty classes with largest train-test performance gap used in the our second experiment. The *debug* column indicates which classes received supervision and are thus debugged.

| # | CLASS | LABEL | DEBUG? | # | CLASS | LABEL | DEBUG? |
|---|---|---|---|---|---|---|---|
| 0 | 001 | Black-footed Albatross | ✓ | 10 | 085 | Horned Lark | |
| 1 | 004 | Groove-billed Ani | | 11 | 086 | Pacific Loon | |
| 2 | 006 | Least Auklet | | 12 | 113 | Baird Sparrow | |
| 3 | 026 | Bronzed Cowbird | | 13 | 122 | Harris Sparrow | |
| 4 | 037 | Acadian Flycatcher | | 14 | 128 | Seaside Sparrow | ✓ |
| 5 | 040 | Olive-sided Flycatcher | | 15 | 137 | Cliff Swallow | ✓ |
| 6 | 043 | Yellow-bellied Flycatcher | ✓ | 16 | 161 | Blue-winged Warbler | |
| 7 | 056 | Pine Grosbeak | | 17 | 183 | Northern Waterthrush | |
| 8 | 070 | Green Violetear | ✓ | 18 | 188 | Pileated Woodpecker | |
| 9 | 076 | Dark-eyed Junco | | 19 | 200 | Common Yellowthroat | |

## D.2 COVID DATA SET

The training set consists of a subset of images retrieved from the GitHub-COVID repository (Cohen et al., 2020) and from the ChestX-ray14 repository (Wang et al., 2017). The number of COVID-negative and COVID-positive radiographs is 7390 and 250, respectively. The test set is the union of PadChest (Bustos et al., 2020) and BIMCV-COVID19+ (Vayá et al., 2020) datasets, totalling 1147 negative and 597 positive images. In (DeGrave et al., 2021), the classifier is trained on 15 classes, i.e., COVID and other 14 pathologies. In (DeGrave et al., 2021), the classifier is trained on a multi-label data set, in which each scan is associated with multiple pathologies, and the evaluation is then performed on COVID positive vs. other pathologies. To simplify the identification and thus the supervision given to the machine, we generated a binary classification problem containing COVID-positive images and images without any pathology. In this setting, supervision is given only on obvious confounders like areas outside the body profile or on arms and elbows. A domain expert can provide detailed supervision on which parts of the lungs the machine must look at.

## E ADDITIONAL RESULTS

**Additional part-prototypes for COVID.** Fig. 9 reports all prototypes for the experiment on the COVID data set.

**Confusion matrices.** The confusion matrix on the left side of Fig. 10 shows that ProtoPNet over-predicts the COVID-positive class, whereas the prediction of ProtoPDebug after three rounds is more balanced since the confounders have been debugged. A domain expert can give additional supervision on which parts of the lungs are relevant for the classification task, further improving the performance.

Figure 9: Four rounds of sequential debugging with ProtoPDebug on COVID. Top row reports the prototypes with non-zero activation: first prototype refers to COVID- and second COVID+. Second row: prototypes with zero activation for each class.

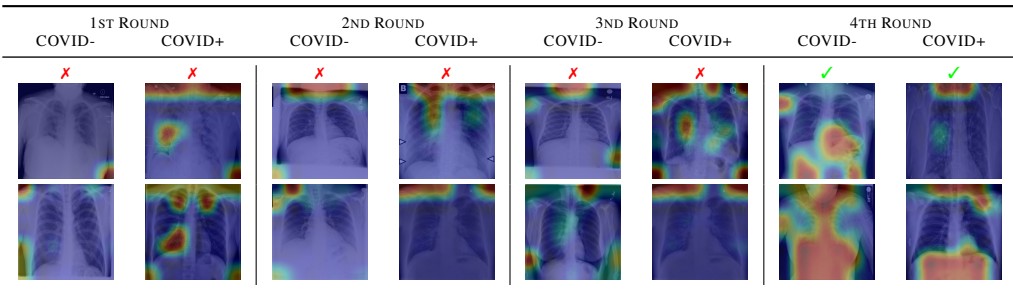

Figure 10: Confusion Matrices on test set where class 0 and 1 is COVID-negative and COVID-positive respectively.

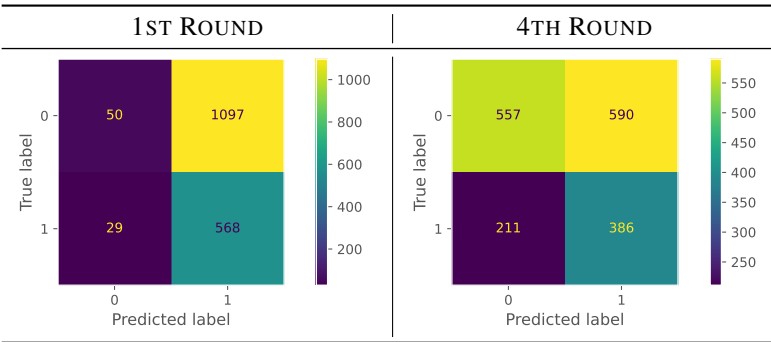

## F  COMPARISON TO REMOVING-AND-FINETUNING

One alternative strategy for debugging ProtoPNets consists of removing confounded part-prototypes (or fixing their weight to zero) and then fine-tuning the aggregation layer. Here, we highlight some limitations of this strategy that ProtoPDebug does not suffer from. In all experiments we use the same setup as in the experiments reported in the main text.

First, this strategy cannot be used when *all* prototypes are confounded. This can occur because, in ProtoPNets, there is no term encouraging the network to learn distinct part-prototypes. In practice, it can happen that *all* part-prototypes learn to recognize the same confound. This is illustrated by the two left images in Fig. 11: here a ProtoPNet trained on the CUB5$_{box}$ data set has learned the same confounder (the colored box) with both of the part-prototypes allocated to the target class. In this situation, the above strategy cannot be applied as there are no unconfounded concepts to rely on for prediction.

What happens if only *some* part-prototypes are confounded? In order to answer this question, we assume that the ProtoPNet loss is augmented with a diversification term $\ell_{\mathrm{div}}$ of the form:

$$\ell_{\mathrm{div}}(\theta) := \frac{1}{|\mathcal{P}|} \sum_y \sum_{\mathbf{p}_{\hat{j}} \in \mathcal{P}^y} \min_{j:\mathbf{p}_j \in \mathcal{P}^y} \|\mathbf{p}_{\hat{j}} - \mathbf{p}_j\|^2. \tag{16}$$

where $\lambda_{\mathrm{div}}$ is a hyper-parameter. This term encourages the different part-prototypes of each class $y$ to be different from each other, and is useful to prevent them to all acquire the same confound.

Now, even if only *some* part-prototypes are confounded, there is no guarantee that the unconfounded ones capture meaningful information. The opposite tends to be true: when a strong confounder is present, the network can (and typically does) achieve high accuracy on the training set by learning to recognize the confounder and allocating all weight to it, and has little incentive to learn other

Figure 11: **Left pair**: prototype activations of a confounded class in CUB5$_{box}$. Notice that both prototypes are confounded. **Right pair**: one prototype activates on the confounder (colored box) whereas the other is neither interpretable nor useful for prediction.

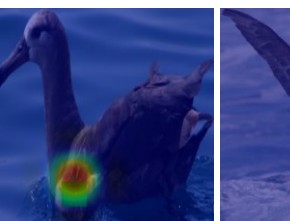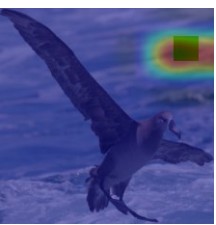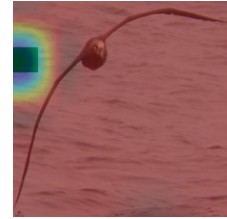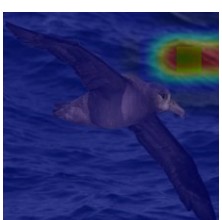

discriminative, high-quality part-prototypes. We validated this empirically by training a ProtoPNet model on the CUB5$_{box}$ data set using the diversification term above, setting $\lambda_{\mathrm{div}}$ to 0.1. The results are shown in Fig. 11 (right): the two activation maps show that one part-prototype activates on the confound, while the other captures irrelevant information and that therefore it would perform very poorly when used for classification.

## G  ABLATING THE REMEMBERING LOSS

We evaluate the contribution of the remembering loss in last debugging rounds of the COVID experiment. The remembering loss avoids performance drop over time. We re-run the last two debugging rounds of the experiment in Figure 5 by setting $\lambda_r$ to zero. In terms of $F_1$, the performance in the third round goes from 0.55 with the remembering loss to 0.44 without. In the fourth round, performance decrease from 0.54 to 0.4.

## H  USER EXPERIMENT

We evaluated the usability of ProtoPDebug in a sequence of debugging round with real users. The participants were asked to review patches of prototype activation on the training images. The patch extraction procedure starts by selecting the five most activated images for each prototype. For each activation map, we generate a binary image in which pixels are set to one if their activation value is at least 95%-percentile of all activation (as in (Chen et al., 2019)), zero otherwise. Then, we extract the contours representing the border of the activation patches. We discard the patches that are too small, difficult to be spotted by the participants and not capturing enough information. We set the threshold to patches having less than 200 pixels area. Finally, we overlay the original images with the extracted patches.

We ran the online experiment through Prolific Academic. For each debugging round experiment, ten participants, laypeople and English native speakers, inspected the images generated as described above in a sequence of questions like Fig. 12. Each experiment took between 9 and 15 minutes according to the number of patches to review. Participants received an hourly fee for their participation. On top of this amount, we gave an additional bonus based on the agreement with the other participants. We randomly selected five images, and we awarded the users whose evaluations agree with at least 80% of others' evaluations for all five images.

We provided the following instruction to the user:

*You will now be presented with a series of images of birds in various natural settings. Each image has a highlighted overlay that goes from dark red (the highlight focus) to light green (the highlight margin).*

*Your task will be to carefully inspect each image and then to determine whether the highlighted overlay covers some part of the bird or whether, instead, it covers exclusively (or very nearly so) the background.*

Table 3 exposes statistics about user feedback over the debugging rounds. The number of patches on which the prototypes activate almost exclusively on the backgrounds decreases by giving more

Figure 12: Question in the user experiment form. The upper part display one activation patch. The user must choose either that the overlay covers part of the bird or exclusively the background.

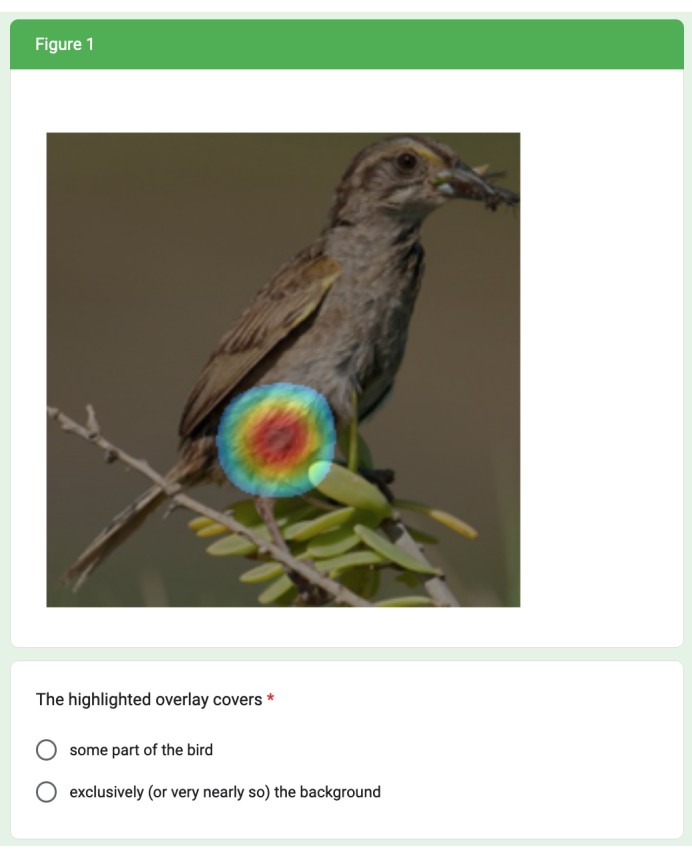

Table 3: Statistics about the patches for all debugging rounds. From left to right: number of patches overlaying the background with the inter-annotator agreement of at least 80%, number of patches on which the agreement is under 80%, number of patches activating on the bird with an agreement of at least 80% and the total number of patches shown to the participants.

| ROUND | # BACKGROUND | # BIRD | # NO AGREEMENT | TOTAL |
|---|---|---|---|---|
| FIRST | 19 | 73 | 5 | 97 |
| SECOND | 6 | 63 | 5 | 74 |
| THIRD | 3 | 73 | 2 | 78 |

supervision to the model. The inter-annotation agreement is 96% and shows that the ProtoPDebug feedback required from the supervisor is simple and effective.

