# OpenReview forum: "Concept-level Debugging of Part-Prototype Networks"
_ICLR.cc/2023/Conference — ICLR 2023 notable top 25%_

### Official Review · Reviewer_wHVs · 2022-10-23

**Confidence:** 4
**Correctness:** 3
**Technical Novelty And Significance:** 4
**Empirical Novelty And Significance:** 4
**Recommendation:** 8

**Clarity, Quality, Novelty And Reproducibility:**

There are some technical choices which I would like to clarify
1. Instead of memorising the re-embedded cut out $x_R$ in Figure 3, why is it not possible to just remember the prototype $p$? If there is high activation, both vectors should be similar.
2. In the constrained distillation loss (11), is it necessary to allow for permutations of the prototypes? I agree that the problem is permutation invariant, but if the weights remain similar, I would expect that the order doesn't change. Here I'm concerned that the enumeration of permutations would create a combinatorial explosion as $k$ increases.
3. In the losses (12) and (13), have you considered the alternative choice of summing the activations (or their opposite) instead of taking the maximum (rest. minimum)? This would be more in line with prior work which does a sum of squares over pixels, see for instance the proof in Appendix A where you have to upper bound the norm by an infinite norm with the usual $\sqrt{k}$ factor. I'm asking this because it isn't so frequent to see $max$ and $min$ directly differentiated for gradient descent and it seems somewhat inefficient. Maybe it would be possible to at least use soft optima?
4. In the first experiment (5.1) I can't find the information of how the annotation is provided to ProtoPDebug? And which percentage of annotated input does it use?

My knowledge of the related literature is not extensive enough to make a definitive assessment of the originality of the method. Based on the provided references the contribution appears novel.

**Strength And Weaknesses:**

This is a well-scoped contribution which is very clearly presented. I find the experimental results particularly convincing of the benefits of ProtoPDebug, especially Figure 4 right which shows a clear gap in terms of generalisation compared to all alternative approaches on the birds dataset. I was able to clearly follow the methodology leading to these results.

I however have some concerns about the main claim that ProtoPDebug is a *"concept-level debugger"*. ProtoPDebug certainly departs from prior work by operating in the embedding space instead of remembering pixel-level masks, but there still seems to be a very strong visual prior throughout the paper:
- I find the user questions (Appendix H) to be heavily biased: *"Your task will be to carefully inspect each image and then to determine whether the highlighted overlay covers some part of the bird or whether, instead, it covers exclusively (or very nearly so) the background."*. This question basically assumes that the confounding factors are already known. To me, this phrasing essentially masks the underlying problem of determining causality, which undermines the claims. It would be better to have a more neutral phrasing, for instance *"Do these patches accurately explain the prediction?"*.
- Section 4 always assume an image structure, such that it is possible to extract a "cut-out", which is then re-embedded into a vector.


**Summary Of The Paper:**

The paper presents a novel interactive learning "debugger" for ProtoPNets. Contrary to the existing IAIA-BL debugger that uses input level annotations (in the form of masks of irrelevant pixels), ProtoPDebug moves towards concept level interaction by storing forbidden and valid concepts in the embedding space, which permits easier transfer of the feedback between inputs. The benefits of this approach are experimentally shown on two different domains: bird classification and COVID diagnostic from chest radiographies. Part of the study is truly interactive, either using a public platform or internally to the authors' research lab.

**Summary Of The Review:**

I find the submission to be a solid and effective contribution. I have some concerns about the generality of the claims but I would still like to recommend acceptance. I'm looking forward to discussion with the authors to clarify some misunderstandings.

---

> ### Author Response · Authors · 2022-11-11
> **Comment to reviewer wHVs**
>
> **Bias in user study**: Our experiment is a first empirical investigation aimed at verifying whether naive users can warn the machine that it is focusing on a confounder and whether the machine can use this feedback to improve its predictions. To do so, we decided to employ a simple but unambiguous question. Accordingly, we excluded all possible questions that required showing participants the prediction of the machine. Indeed, the correct vs. incorrect prediction would have easily interacted with the evaluation of the informativeness of the patch (e.g., it is not obvious what a user should answer when the prediction is wrong but the patch is not confounded). We were fully aware that the specific question that we used assumes that the confounding factor (namely the background) is already known, and therefore that users’ answers would have allowed us to debug only it. However, this was enough to obtain an increase in performance. As a matter of principle, the use of a more general question that allows debugging other confounders, should increase it even more. This further step, though, would probably require involving domain experts, who share a background knowledge that allows them to interpret this question effectively and coherently. Indeed, the answers to a question that asks, for example, to decide if a patch is informative in order to classify a specific image would be likely influenced by users’ interpretation of what 'informative' means in the specific context at issue (i.e., a patch overlaying the bird’s chest is informative because it is not on the background or because this part of the bird can be used to distinguish it from other species?). Future research could explore this topic in more detail by empirically comparing the effectiveness of different questions for eliciting users’ feedback.
>
> We’d also like to mention that one of the authors, who lead the design of this experiment, is a cognitive psychologist with extensive experience in eliciting human judgement.
>
> **Assuming cut-out makes sense**: ProtoPDebug is meant to be applied in settings where ProtoPNets are a reasonable choice, such as image classification tasks in which instances can be classified based on presence/absence of concrete elements captured by part-prototypes.  In these settings, cut-outs exemplify (to the user, and as feedback to the machine) what the part-prototypes activate on.  This seems like a reasonable assumption to us. Extensions to, e.g., textual data are left to future works;  however, we point out that - as mentioned in the Related Work - ProtoPDebug should apply essentially as-is also to iProtoTREX, a generalization of ProtoPNets designed for NLP tasks.
>
> **Memorizing the embedding rather than the cut-out**: This strategy unfortunately does not work: fine-tuning the model involves updating the embedding layers, meaning that any previously memorized embeddings may lose all meaning upon fine-tuning.  Using them to constrain the model could actually be detrimental for recognition performance.  We will clarify this in the paper.
>
> **Permuting the part-prototypes**: There is no combinatorial explosion - the ordering is done implicitly by looking up the closest part-prototype in $\mathcal{F}$ or $\mathcal{V}$ in Eqs. 12 and 13.
>
> **Replace min/max with sum aggregator**: Good point. It is entirely possible that the sum aggregator yields better models - as it would distribute the loss over all learned part-prototypes. It may be worth evaluating this empirically. We did not try this as it was not necessary in our experiments.
>
> **Annotations in Section 5.1**: This first experiment evaluates a single debugging round and reports how performance changes across fine-tuning epochs. All supervision is made available to ProtoPDebug from the beginning. Specifically, the set F includes one cut-out for each colored box added to the dataset (i.e., three cut-out representing the colored boxes), which mimics the feedback that an informed user would give.

---

> > ### Comment · Reviewer_wHVs · 2022-11-12
> > **Answer to comments**
> >
> > **Bias in user study:** I really didn't mean to question the authors' ability to design a good experimental study, and I apologise if it was understood in this way. By bias, I was simply emitting concerns about how much domain knowledge was implanted in the phrasing of the user study. I understand your answer, and I don't think that the assumption you made is bad. However I think it raises some questions about generality, in particular the claims that ProtoPDebug is fully a concept-level debugger and that it is always accessible to laypeople.
> >
> > **Assuming cut-out makes sense:** This would then support my point that some assumptions are specific to the visual domain?
> >
> > **Memorizing the embedding rather than the cut-out:** thank you for clarifying my misunderstanding.
> >
> > **Permuting the part-prototypes:** my question was related to equation (11) (the $\min_{\pi}$ term), not to (12) and (13).
> >
> > **Replace min/max with sum aggregator:** thank you for these additional details.
> >
> > **Annotations in Section 5.1:** I suppose this is what was meant by *"We compare ProtoPDebug, with supervision on a single instance per confounder"*. Could you please add to the paper the information that *"All supervision is made available to ProtoPDebug from the beginning."*?
> >
> >
> > Based on the other very positive reviews and the author feedback I would like to reaffirm my acceptance recommendation with more confidence. **I would increase my rating to 7 if this was allowed by the form.**

---

> > > ### Author Response · Authors · 2022-11-15
> > > **Answer**
> > >
> > > We really appreciate you spending the effort to interact with us and provide valuable, constructive feedback about our paper.
> > >
> > > **Generality of ProtoPDebug**:  Thank you for the clarification.  It is true that we focused on confounds that can be diagnosed by visually inspecting individual part-prototypes (over time), chiefly due to our dependency from ProtoPNets.  One side benefit of this restriction is it facilitates understanding and thus interaction even with non domain experts - although, naturally, some expertise on the user’s end is always assumed.
> > >
> > > At its core, our assumption is that confounded behavior can be identified through explanations, an assumption that underlies all approaches to explanation-based debugging. We agree that, depending on the type of explanations used some confounds may be undetectable.  Thanks to ProtoPNets, we can actually build on explanations (including part-prototype weights, mappings to training examples, and saliency maps) that are strictly more expressive and specific than those used in alternative approaches (such as saliency maps in RRR and IAIA-BL).  We also agree that confounds that do not affect part-prototypes cannot be detected - however, they are also less likely to affect predictions, because ProtoPNets derive their predictions from the part-prototypes only.
> > >
> > > A more serious issue is that if the data set used for debugging contains too few examples, explanations may fail to capture buggy behavior altogether.  This, however, could be addressed independently of ProtoPDebug by injecting more (representative) examples in said data set.
> > >
> > > Concerning the role of erroneous predictions in eliciting feedback: one solution to avoid any confusion on the user’s end would be to implement a two-stage elicitation strategy whereby users first determine whether a prediction is indeed correct and, conditionally on this being the case, provide feedback on the part-prototypes.  I.e., this more complex type of feedback would be only elicited in the “right for the wrong reasons” case (see [Teso et al., 2019]).  This would be a simple-to-implement but useful addition to ProtoPDebug, and we plan to evaluate it in future work.
> > >
> > > We will clarify these points in the “Benefits and Limitations” paragraph of the paper.
> > >
> > > **Permuting the part-prototypes**: Eq. (11) is introduced to help build intuition about the forgetting and remembering losses, but it is never instantiated in practice.  Our implementation relies solely on Eqs. (12) and (13) to fine-tune the model, as shown in Algorithm 1.  We will rework the text to make this more clear.
> > >
> > > **Annotations in Section 5.1**: Definitely.  We updated the last sentence of that paragraph to read “In this setting, all supervision is made available to all methods from the beginning, and [...]”.

---

> > > > ### Comment · Reviewer_wHVs · 2022-11-15
> > > > **Answer**
> > > >
> > > > Thank you for your answer that was simultaneous to my decision to increase the score. I appreciate the prospective clarification of "Benefits and Limitations" to reflect as much of the discussion as possible.
> > > >
> > > > The assumption that *"confounded behavior can be identified through explanations"* seems very reasonable because I don't perceive so much difference between a cause (in the sense of causality) and an explanation. It seems that the main challenges of debugging are (i) enlarging the class of supported explanations. (ii) diminishing the sample complexity by bringing more generalisation of the user feedback. I can see that your submission is making substantial progress in both directions and that is definitely a very valuable contribution. Perhaps a title like "Towards Concept-Level Debugging of Part-Prototype Networks" would be even more adapted but I agree that it's a nitpick and that it should be up to you to decide.

---

> > > ### Comment · Reviewer_wHVs · 2022-11-15
> > > **Score increase**
> > >
> > > I have been thinking some more about this submission in the last few days and despite persistent concerns about the "concept-level" claim, I acknowledge that there are some really strong aspects to the paper which deserve to be highlighted. **I have increased my score to 8.**

---

### Official Review · Reviewer_oazP · 2022-10-24

**Confidence:** 4
**Clarity, Quality, Novelty And Reproducibility:** This work is strong in terms of clari…
**Correctness:** 3
**Technical Novelty And Significance:** 4
**Empirical Novelty And Significance:** 4
**Recommendation:** 8

**Details Of Ethics Concerns:**

There are no ethical concerns.

**Strength And Weaknesses:**

Strengths:

- Novelty: The concept-level debugger, ProtoPDebug, is a very novel idea. It allows us to interactively debug a ProtoPNet, without requiring instance-level fine-grained annotations.
- Technical soundness: ProtoPDebug is in general technically sound. The idea of fine-tuning the network using a loss that discourages new prototypes to activate on forbidden concepts and encourages them to activate on valid concepts is very neat.
- Writing: The paper is clearly written and easy to follow.

Weaknesses:

- Minor issue: I have a minor issue with equation (11). In particular, equation (11) introduces a re-ordering on part prototypes of the updated model. Could you clarify what is meant by "re-ordering the part-prototypes"? Is \pi a permutation of the part prototypes of the updated model? If so, shouldn't the corresponding weight entries in w' be re-ordered as well? Please explain.

**Summary Of The Paper:**

In this paper, the authors proposed a concept-level debugger, called ProtoPDebug, for prototypical part networks (ProtoPNets). ProtoPDebug addresses the issue of confounding prototypes learned by a ProtoPNet, by performing human-in-the-loop model debugging. In each debugging iteration, a human expert can mark a prototype as a "confounder" and add the corresponding highly activated region from the prototype's source image to a set of "forbidden concepts" F. Optionally, a human expert can also mark a prototype as a "high-quality" prototype and add the corresponding highly activated region from the prototype's source image to a set of "valid concepts" V. ProtoPDebug can then fine-tune the network by minimizing a loss that encourages the new parameters to be close to the existing ones, and encourages the new prototypes to not activate on the set of forbidden concepts and to activate on the set of valid concepts. The key benefit of ProtoPDebug (over an instance-level debugger such as the IAIA-BL) is that ProtoPDebug does not require fine-grained annotations on individual input instances at all, and is therefore easy to use.

**Summary Of The Review:**

Based on the strengths of this paper, I recommend to accept the paper.

---

> ### Author Response · Authors · 2022-11-11
> **Comment to reviewer oazP**
>
> Thank you for your positive feedback.
>
> **Equation 11**: Our intuition is that the order in which part-prototypes are learned is insubstantial:  it does not matter whether a (possibly confounded) part-prototype appears first, second, or last in the list of learned part-prototypes.  The permutation simply matches each learned part-prototype **p** to the “best matching” part-prototype in the original model.
>
> In practice, the loss we use to fine-tune/debug the model takes permutation-invariance into account by matching each learned part-prototype to the “closest” element in the forbidden set, see Eqs. 12 and 13.  In other words, the loss in Eq. 12 penalizes part-prototype activation without explicitly rearranging the learned part-prototypes.
>
> As a result , the part-prototypes that activate the most on the forbidden patterns end up changing upon fine-tuning, and so will their associated weights.  Eq. 13 works similarly, but it incentivizes at least one learned part-prototype to activate on cut-outs that the user deems useful.

---

> > ### Comment · Reviewer_oazP · 2022-11-21
> > **Thank you for the explanation**
> >
> > Thank you for the explanation! Good work!

---

### Official Review · Reviewer_FdCM · 2022-10-24

**Confidence:** 5
**Correctness:** 3
**Technical Novelty And Significance:** 3
**Empirical Novelty And Significance:** 3
**Recommendation:** 8

**Clarity, Quality, Novelty And Reproducibility:**

+ clear paper description
+ incremental novelty but crucial for moving this area forward

**Strength And Weaknesses:**

+ Interesting idea of iterative explanation
- Missing larger scale end-user evaluation

**Summary Of The Paper:**

This paper presents a concept-level debugger for ProtoPNets in which a human supervisor, guided by the model’s explanations, supplies feedback in the form of what part-prototypes must be forgotten or kept.

**Summary Of The Review:**

Interesting paper presenting an approach to refine explanation through user interaction. I really think this approach has some added value, and is an interesting way of tackling (incrementally) the problem of eplainability.

---

> ### Author Response · Authors · 2022-11-11
> **Comment to reviewer FdCM**
>
> Thank you for your encouraging feedback.
>
> **Larger-scale evaluation**: Our evaluation considers a total of 10 users for several interaction rounds, and follows a principled evaluation protocol. This gives us confidence that ProtoPDebug works as intended and that it can be used consistently by non-expert users.  Notice also that inter-annotator agreement was very high (96%, see Section 5.2). Regardless, we are definitely interested in carrying out further studies in the context of a high-stakes applications (e.g., medical diagnosis).  However, we will leave this to future work.

---

### Official Review · Reviewer_tqKr · 2022-10-27

**Confidence:** 3
**Correctness:** 3
**Technical Novelty And Significance:** 3
**Empirical Novelty And Significance:** 3
**Recommendation:** 8

**Clarity, Quality, Novelty And Reproducibility:**

Clarity: Adequate
Quality: High
Novelty: Adequate
Reproducibility: Adequate.

**Strength And Weaknesses:**

Strengths
- The paper makes good progress on debugging models using concepts. The authors write well: this makes it quite straightforward to see how this paper varies from existing work. The contributions are thus clear.
- The layperson validation of the method is quite convincing and shows the utility of ProtoPDebug.
- Figure 3, which should be Algorithm 1(?), is straightforward to implement

Weaknesses
- Please explain what is meant by "θ' is consistent with F" in Equation 10
- It would be nice to make the the debugging process iterative (instead of one off to build up V or F); e.g., how many examples does it take before no debugging is needed? How much does debugging vary between participants

**Summary Of The Paper:**

This paper proposes ProtoPNets, a method to debug models at the concept level.

**Summary Of The Review:**

This is a strong paper that proposes an effective concept-level debugger that the authors validate not only using quantitative experiments but also using human-subject experiments. I foresee this paper garnering some attention for its simplicity and clear validation.

---

> ### Author Response · Authors · 2022-11-11
> **Comment to reviewer tqKr**
>
> Thank you for your encouraging review.
>
> **Equation 10**: By “θ is consistent with F” we mean that the ProtoPNet parameterized by θ should not activate highly, if at all, with the part-prototypes contained in F.  In other words, we view the user’s feedback as constraints on the model’s behavior.  We have rephrased the text to make this clearer.
>
> **Making the debugging process iterative**: Allowing for multiple rounds of correction within a single round of debugging is a very interesting research direction - particularly for dealing with bugs that have complex causes, e.g., in hierarchical classification tasks - and, as a matter of fact, we are actively pursuing it in a separate work. Since here we are dealing with multi-class classification, we restricted ourselves to this simpler (but non-trivial) case.
>
> Notice that, despite this restriction, in ProtoPDebug the user can still provide feedback on multiple part-prototypes per debugging round, and that - over time - they can also address bugs that the model might unintentionally acquire as a consequence of fine-tuning.
>
> Concerning termination, for now debugging stops once the user is satisfied with the model’s predictions and explanations over the examples on which each prototype activates the most.  Of course, this does not automatically guarantee the model to be completely unconfounded.  However, it is easy for users to run ProtoPDebug on-demand when they spot new cases of misbehavior.
>
> Variance across participants: In the user study, feedback was very consistent among participants, yielding a 96% inter-annotation agreement, as we mentioned in the Results paragraph of Section 5.2, p 8.
>
> **Figure 3**: Thank you for pointing this out.  We have renamed it to Algorithm 1 for clarity.

---

### Decision · Program_Chairs · 2023-01-20

**Decision:**

Accept: notable-top-25%

**Justification For Why Not Higher Score:**

This paper could be considered for an oral.

**Justification For Why Not Lower Score:**

This is a strong paper, with ideas that may be more generally applicable.

**Metareview: Summary, Strengths And Weaknesses:**

This paper presents a "debugger", which is a method for human experts to provide feedback on model predictions, specifically on what portion of the input is relevant, which is then further used to finetune the model. The reviewers agreed that that this is a creative new contribution in an important area, which is relevant both for work in interpretability and in prototype-based learning.

**Note From Pc:**

if the above contains the word "oral" or "spotlight" please see: "oral" presentation means -> notable-top-5% and "spotlight" means -> notable-top-25%. As stated in our emails, we are disassociating presentation type from AC recommendations